# N^6^-Methyladenosine RNA Modification Regulates the Differential Muscle Development in Large White and Ningxiang Pigs

**DOI:** 10.3390/cells13201744

**Published:** 2024-10-21

**Authors:** Hao Gu, Kang Xu, Zhao Yu, Zufeng Ren, Fan Chen, Changfan Zhou, Wei Zeng, Hongyan Ren, Yulong Yin, Yanzhen Bi

**Affiliations:** 1Key Laboratory of Animal Embryo Engineering and Molecular Breeding of Hubei Province, Institute of Animal Sciences and Veterinary Medicine, Hubei Academy of Agricultural Sciences, Wuhan 430070, China; guhao@hbaas.com (H.G.); yuzhao0529@163.com (Z.Y.); 5025144800@163.com (Z.R.); fanchen@hbaas.com (F.C.); zhouchangfan@webmail.hzau.edu.cn (C.Z.); zw8023dfl@hbaas.com (W.Z.); 18971609780@163.com (H.R.); 2Institute of Subtropical Agriculture, Chinese Academy of Sciences, Changsha 410125, China; xukang2020@163.com; 3Hubei Hongshan Laboratory, Wuhan 430070, China

**Keywords:** pig muscle, m^6^A, MeRIP-seq, WFS1, YTHDF2

## Abstract

N6-methyladenosine (m^6^A) is the most common modification in eukaryotic RNAs. Growing research indicates that m^6^A methylation is crucial for a multitude of biological processes. However, research on the m^6^A modifications in the regulation of porcine muscle growth is lacking. In this study, we identified differentially expressed genes in the neonatal period of muscle development between Large White (LW) and NingXiang (NX) pigs and further reported m^6^A methylation patterns via MeRIP-seq. We found that m^6^A modification regulates muscle cell development, myofibrils, cell cycle, and phosphatase regulator activity during the neonatal phase of muscle development. Interestingly, differentially expressed genes in LW and NX pigs were mainly enriched in pathways involved in protein synthesis. Furthermore, we performed a conjoint analysis of MeRIP-seq and RNA-seq data and identified 27 differentially expressed and m^6^A-modified genes. Notably, a typical muscle-specific envelope transmembrane protein, WFS1, was differentially regulated by m^6^A modifications in LW and NX pigs. We further revealed that the m^6^A modification accelerated the degradation of WFS1 in a YTHDF2-dependent manner. Noteworthy, we identified a single nucleotide polymorphism (C21551T) within the last exon of WFS1 that resulted in variable m^6^A methylation, contributing to the differing WFS1 expression levels observed in LW and NX pigs. Our study conducted a comprehensive analysis of the m^6^A modification on NX and LW pigs during the neonatal period of muscle development, and elucidated the mechanism by which m^6^A regulates the differential expression of WFS1 in the two breeds.

## 1. Introduction

Pigs serve as a major source of meat and an ideal model for human diseases, making the study of muscle development crucial for both agriculture and biomedicine [1]. The pronounced phenotypic contrasts between commercial breeds (lean type), known for their rapid muscle growth, and indigenous breeds (obese type), which exhibit slower growth rates, provide an exceptional model for identifying key genes and pathways involved in muscle development [2,3]. Whole-genome re-sequencing of the skeletal muscle of Landrace and Tongcheng pigs at 27 time points from embryonic to adult further elucidated the phased nature of muscle development and the notable differences in muscle growth between lean- and obese-type pigs [4]. Although extensive sequencing has been performed at the transcriptional level in porcine muscle, high-throughput sequencing studies targeting transcriptional regulation, especially epigenetic modifications, remain scarce.

m^6^A is the most abundant post-transcriptional modification of eukaryotic mRNA, which regulates almost all aspects of mRNA metabolism, from transcription and precursor mRNA processing to translation and degradation [5,6]. m^6^A modification is catalysed by a complex of ‘writer’ enzymes, including methyltransferase-like 3 (METTL3), METTL14, Wilms tumour-associated protein (WTAP), and ZC3H13 [7,8]. METTL3 functions as the enzyme’s catalytic component, while METTL14 provides the RNA-binding capability [9]. These two proteins are conserved across mammalian species and together they constitute a robust heterodimeric structure [10,11]. m^6^A modification can be dynamically reversed by ‘eraser’ enzymes like FTO and ALKBH5. ALKBH5 regulates mRNA splicing and stabilisation in the nuclei of spermatogonia and round spermatocytes by removing m^6^A from precursor mRNAs and mediates the production of mRNAs with a longer 3′UTR [12,13]. FTO is more widely expressed, and its potent and prevalent m^6^A demethylation exhibits promotional effects in a variety of cancers. FTO regulates mouse preadipocyte differentiation by modulating the alternative splicing of mRNAs [14]. Methylation reading proteins, referred to as ‘readers’, recognise and bind to m^6^A sites. The diversity of these readers dictates the various regulatory roles governed by m^6^A modifications. For example, m^6^A is involved in pathways mediated by METTL3, ALKBH5, and YTHDC1, which regulates mRNA export from the nucleus to the cytoplasm [12,15]. Multiple mechanisms of m^6^A promoting mRNA translation have been confirmed, including the YTHDF1-eIF3 pathway, cap-independent translation, and IGF2BP-mediated translation [16,17,18]. In terms of mRNA stability control, early studies generally believed that m^6^A was an unstable factor that promoted mRNA degradation mainly through YTHDF2. Recent findings have uncovered an alternative role for m^6^A, where IGF2BP proteins mediate its function to enhance mRNA stability and retention [18,19].

The growing body of research highlights the pivotal regulatory function of m^6^A modification in porcine muscle development, with essential pathways like p38MAPK, mTOR-PGC1α, MSTN-SMAD3, and P14K-Akt having been identified as targets of m^6^A modulation [20,21,22,23]. Although methylated-RNA immunoprecipitation sequencing (MeRIP-seq) has facilitated the mapping of the m^6^A regulatory landscape and the discovery of m^6^A-regulated genes in pig muscle [24,25], studies concentrating on the neonatal stages of muscle development are limited. Notably, differences in m^6^A modifications between lean and obese pig breeds during muscle development have not been elucidated.

In this study, we meticulously analysed the dynamic m^6^A methylome and delineated the distinctions in the methylome and transcriptome of skeletal muscle between indigenous Chinese NX pigs and commercial LW pigs. Multi-omics analysis of m^6^A and experimental data indicated that m^6^A modification promotes degradation of the WFS1 gene by binding with the reader protein YTHDF2. Noteworthy, a single nucleotide polymorphism (SNP) within the conserved m^6^A binding motif influences variability in WFS1 expression in NX and LW pigs. Our findings provide novel insights into the role of epigenetic modifications in muscle development.

## 2. Materials and Methods

### 2.1. Animals

Three five-day-old NX and three five-day-old LW pigs were obtained from Hunan Chuweixiang Agriculture and Animal Farming Co., Ltd. (Changsha, China). To rule out the potential influence of gender, male piglets were selected for our study. All animal experiments were conducted in accordance with the National Research Council Guide for Care and Use of Laboratory Animals, also approved by the Animal Care and Use Committee of the Hubei Academy of Agricultural Sciences (HBAAS-2023-014).

### 2.2. RNA Extraction and m^6^A MeRIP

Total RNA was extracted from samples using TRIzol reagent. For MeRIP, total RNA was fragmented at 70 °C for 10 min using an RNA fragmentation buffer. The fragmented RNA was ethanol precipitated and then incubated with m^6^A antibody-conjugated magnetic beads at 4 °C for 2 h. After washing, the beads were resuspended in an IP reaction mixture. The m^6^A-RNA was immunoprecipitated, washed three times with an IP buffer, and eluted in an m^6^A competitive elution buffer with shaking for 1 h. The eluted m^6^A-RNA was then purified using a phenol—chloroform—isoamyl alcohol extraction.

### 2.3. Library Preparation

The MeRIP library preparation was executed with the extracted m^6^A RNA, adhering to the guidelines of the SMARTer stranded total RNA-Seq kit version 2 (Takara/Clontech). Initially, we directly converted both m^6^A-enriched and input RNA samples into the first-strand cDNA, omitting the fragmentation process. We then adhered to the remaining steps of the SMARTer stranded total RNA-Seq kit version 2 protocol. The IP RNA libraries were PCR amplified for up to 16 cycles, and the input libraries were amplified for no more than 12 cycles. Subsequently, each library’s quality was assessed with an Agilent 2100 Bioanalyzer, and their concentrations were determined by real-time PCR prior to initiating the sequencing process.

### 2.4. Sequencing and Analysis

After confirming the quality of the libraries, they were pooled according to the effective concentration of the libraries and the amount of target output data required and sequenced on an Illumina NovaSeq platform using the paired-end 150 (PE150) sequencing strategy. Subsequently, the filtered data were aligned to the reference genome using the hisat2 analyser.

### 2.5. Differential Expression Analysis

Differential expression analysis between the NX and LW groups was conducted using edgeR package Genes with an adjusted *p*-value of less than 0.05 and an absolute log2 (fold change) greater than 1 were classified as differentially expressed.

### 2.6. MeRIP-Seq Analysis

Sequencing data were aligned to the reference genome using HISAT2. Enrichment regions of reads in the IP samples relative to the input were identified using exomePeak package. Subsequently, the detected peaks were annotated with the ChIPseeker R package. Differential m^6^A modifications were identified using the exomePeak R package, and statistical analysis of the differential peaks was performed using the ChIPseeker R package.

### 2.7. RNA m^6^A Dot Blot Assay

The total RNA was denatured at 95 °C for 5 min and immediately chilled on ice for 5 min. Next, 200 ng of RNA was spotted onto nylon membranes, air-dried, and crosslinked with UV light for 1 min. After thrice 5 min TBST washing, the membranes were stained with methylene blue and destained. For the analysis, an m^6^A-specific antibody (1:1000, ABclonal, Wuhan, China) was applied, followed by a goat anti-rabbit IgG secondary antibody (1:5000, ABclonal). Membranes were imaged after ECL application, and m^6^A levels were analysed using the ImageJ (NIH, Bethesda, MD, USA) analyser.

### 2.8. Histology Staining

The muscle tissues were cut into 4 μm thin slices. Immunofluorescence staining was carried out following established procedures [26]. The following antibodies were used: anti-laminin (1:200; Santa Cruz Biotechnology, CA, USA), MyH7 (1:200; Santa Cruz Biotechnology), Pax7 (1:100; Santa Cruz Biotechnology), Ki67 (1:100; Santa Cruz Biotechnology), and anti-FTO (1:100; ABclonal). Images were captured using an optical microscope (BX53; OLYMPUS, Tokyo, Japan) and analysed using an ImageJ analyser.

### 2.9. Porcine Skeletal Muscle Satellite Cell Isolation and Culture

Primary intramuscular pre-adipocytes were isolated from the longissimus dorsi of three-day-old newborn piglets. The muscles of euthanised gilts were harvested, rinsed with PBS containing 1% penicillin–streptomycin, and minced before enzymatic digestion in 2 mg/mL collagenase I at 37 °C with agitation for 2–3 h. The digested tissue was sequentially filtered through sieves to obtain cell suspensions. This suspension was subjected to differential adhesion culture in enriched RPMI 1640 medium supplemented with 20% FBS, 1% non-essential amino acids, 1% chicken embryo extract, and 4 ng/mL basic fibroblast growth factor to isolate the muscle cells. After 2 h, the supernatant containing the purified muscle cells was transferred to a collagen-coated cell culture flask for proliferation at 37 °C in a 5% CO_2_ environment.

### 2.10. Western Blotting

Total protein extraction was conducted from tissue or cell samples using a RIPA buffer, enhanced with 1% PMSF (Beyotime, Shanghai, China). The extracted proteins were denatured at 95 °C for 5 min in a 6× SDS-PAGE loading buffer. Western blotting was performed according to previously established protocols. The following antibodies were used: anti-FTO (1:1000; ABclonal), METTL14 (1:200; Santa Cruz Biotechnology), anti-MyHC (1:200; Santa Cruz Biotechnology), WFS1 (1:1000; Abcam, Cambridge, UK), and anti-GAPDH (1:2000; BOSTER, Wuhan, China).

### 2.11. Cell Immunofluorescence Staining

PSCs were first fixed utilizing a 4% paraformaldehyde solution. Subsequently, cell permeabilisation was achieved with a 0.5% Triton X-100 solution for a duration of 30 min. Once permeabilisation was complete, the cells underwent three washes to remove any residual Triton X-100. Thereafter, the cells were incubated with QuickBlock™ blocking buffer for 2h to prevent non-specific antibody binding. Subsequently, they were incubated with a primary MyHC antibody (Santa Cruz Biotechnology) overnight, followed by a 1h incubation with a secondary antibody. Nuclei were visualised by DAPI staining. Images from at least three random fields were obtained using an optical microscope (BX53; OLYMPUS) and analysed using an ImageJ analyser for quantification.

### 2.12. RNA Immunoprecipitation qPCR

RIP was performed following the protocol provided with the Magna RIP kit (Millipore, Billerica, MA, USA). The m^6^A-RIP assay was performed using approximately 1 × 10^7^ PSCs. The PSCs were centrifuged, re-suspended in 100 μL of RIP lysis buffer containing protease and RNase inhibitors, and lysed. The lysates were then incubated with anti-m^6^A antibody or IgG control antibody overnight at 4 °C. Beads coated with the respective antibodies were added and the mixture was incubated to facilitate RNA immunoprecipitation. The captured RNA was purified using the RNeasy MinElute Cleanup Kit (Qiagen, Hilden, Germany) and transcribed into cDNA using the PrimeScript RT Master Mix (TaKaRa, Tokyo, Japan). Gene expression levels were quantified by qPCR.

### 2.13. Real-Time PCR Analysis

Total RNA was extracted using TRIzol reagent (Invitrogen, Carlsbad, CA, USA) following the manufacturer’s guidelines. The SYBR qPCR Mix (Toyobo, Tokyo, Japan) was utilised for real-time PCR on a QuantStudio 1 instrument. The primer sequences are listed in Appendix A. Relative RNA expression levels were determined employing the Ct (2^−ΔΔCt^) method.

### 2.14. siRNA Synthesis and Cell Transfection

siRNAs for FTO, YTHDF2, and the negative control (NC) were designed and synthesised by GenePharma (Shanghai, China). The siRNA oligonucleotide sequences were as follows:

FTO (sense 5′-GCACCUACAAGUACCUGAATT′);

YTHDF2 (sense 5′-GAGACUGGAUGCUGCUUAUTT′);

NC (sense 5′-UUCUCCGAACGUGUCACGUTT-3′).

PSCs were transfected with siRNA oligonucleotides using Lipofectamine 3000 (Invitrogen), following the manufacturer’s guidelines.

### 2.15. Luciferase Assay

A dual-luciferase activity assay was performed using the Promega dual-luciferase reporter assay system. We diluted the 5× passive lysis buffer and luciferase assay substrate with appropriate diluents and prepared a 50× stop substrate in a stop buffer. Thirty-six hours after transfection with the luciferase reporter plasmids, the cells were washed and lysed with passive lysis buffer and luciferase activity was measured using a 2030 Multilabel Reader VICTOR×2. Initially, 10 μL of cell lysate was mixed with LAR II, and firefly luciferase activity (RLU1) was detected. Subsequently, Stop&Glo was added to quench firefly luciferase and initiate Renilla luciferase activity (RLU2), with the ratio RLU2/RLU1 indicating the relative luminescence intensity and trend.

## 3. Results

### 3.1. Differences between LW and NX Pigs in the Neonatal Stages of Muscle Development

We investigated the characteristics of the longissimus dorsi muscle in five-day-old NX and LW pigs. Immunofluorescence staining for lamins indicated that muscle fibres in NX pigs were sparsely distributed, with a greater number of smaller fibres and fewer larger fibres, leading to a smaller average cross-sectional area (Figure 1A,C). We evaluated the growth rate of muscle fibres by statistically analysing the ratio of cell nuclei to muscle fibres, and found that the ratio in NX pigs was significantly lower than that in LW pigs (Figure 1B). The presence of oxidative muscle fibres was assessed using immunofluorescence staining for MYH7, revealing that LW pigs had a higher proportion of these fibres, with LW pigs at 26.5% and NX pigs at 12.3% (Figure 1D). To evaluate the differentiation and proliferation capacities of the muscle cells, we performed immunofluorescence staining for PAX7 and KI67. The results indicated that both PAX7^+^ and KI67^+^ cells were more abundant in the muscles of LW pigs, with KI67^+^ cells at 22.5% in LW pigs and 15.86% in NX pigs, and PAX7^+^ cells at 18.5% in LW pigs and 9.6% in NX pigs (Figure 1E,F). Overall, muscle development in NX and LW pigs showed great differences at the neonatal stage, with LW pigs exhibiting greater proliferative and differentiative capacities. These differences are beneficial to our subsequent studies on the regulation of muscle development.

### 3.2. m^6^A Podifications Regulated Porcine Muscle Development

To elucidate the role of m^6^A modification in skeletal muscle development, we conducted a comprehensive analysis of mRNA and protein expression levels of m^6^A ‘writer’ genes (METTL3, METTL14, and WTAP) and ‘eraser’ genes (FTO and ALKBH5) in the longissimus dorsi muscle of LW and NX pigs (Figure 2A). We observed higher levels of demethylases and lower levels of methylases in LW pig muscles (Figure 2B,C). Consistent with this finding, the global RNA m^6^A levels detected by the m^6^A dot blot assay were also significantly lower in the muscles of LW pigs (Figure 2D). To verify the relationship between m^6^A modification and myogenic differentiation, we measured FTO expression during PSC differentiation. The protein level of FTO increased along with the differentiation of PSCs myoblasts (Figure 2E). Subsequently, we examined whether the changes in global m^6^A levels affected myogenic differentiation by silencing or overexpressing FTO (Appendix A). During PSCs differentiation, FTO overexpression enhanced myotube formation, whereas si-FTO significantly inhibited this process (Figure 2F).

### 3.3. Global Features of mRNA m^6^A Modification in LW and NX Porcine Muscles

To gain insight into the identity of m^6^A-modified mRNAs involved in muscle development in LW and NX pigs, we performed m^6^A RNA immunoprecipitation followed by sequencing (meRIP-seq) of the longissimus dorsi of LW and NX pigs (Figure 3A). We obtained an average of 52,177,027 correctly paired map reads per sample, with a mapping rate greater than 90%. In total, we identified 2373 and 2122 m^6^A peaks in the LW and NX muscles, respectively (Figure 3B). We then analysed the binding positions of these m^6^A peaks on the genome and found that these peaks were highly enriched near stop codons and 3′UTRs (Figure 3C). We used the HOMER analyser to identify the m^6^A motif, and the typical m^6^A motif RRACH (R=G or A and H=A, C, or U) was enriched in both the LW and NX groups (Figure 3D). Analysis of the m^6^A distribution at different chromosomal loci revealed that the highest frequency of m^6^A methylation was found on chromosome 1, and there was no significant difference in the distribution of m^6^A peaks on the LW and NX chromosomes (Appendix A). GO enrichment analysis showed that most of the m^6^A-modified genes in muscle were significantly enriched in terms of muscle cell development, myofibrils, cell cycle, and ‘phosphatase regulator activity’ (Figure 3F). KEGG enrichment analysis showed that m^6^A modification is involved in the regulation of a variety of pathways with broad impacts, including the TGF-β, PPAR, AMPK, and MAPK signalling pathways (Figure 3G).

### 3.4. Analysis of Differential m^6^A Modification Genes in NX and LW Pigs

Differential m^6^A modification genes were identified by *p* < 0.05 and fold change ≥ 2, and a total of 546 were identified, of which 246 were high m^6^A modification and 300 low modifications in the NX porcine muscle (Figure 4A). The distribution of these differentially modified genes on chromosomes is shown in Figure 4B. We analysed the genomic distribution of differential m^6^A modification sites and found these sites were enriched at the last exon rather than the 3′ UTR region (Figure 4C,D). Among genes with increased m^6^A levels in LW compared with NX, there was significant enrichment of the ‘haematopoietic cell lineage’, ‘T cell receptor signalling pathway’, and ‘PI3K-Akt signalling pathway’ (Figure 4E). In contrast, pathways including the Wnt, thyroid hormone, and cAMP signalling pathways were enriched among genes with decreased m^6^A levels (Figure 4F).

### 3.5. Analysis of Differentially Expressed Genes in NX and LW Pigs

Using the input data, we generated RNA-seq results for muscle tissues from NX and LW pigs. By analysing the RNA-seq results using edgeR software, we identified 1208 differentially expressed genes (DEGs), of which 493 were up-regulated and 715 were down-regulated in the NX group (Figure 5A). A hierarchical clustering analysis revealed a significant difference between the NX and LW groups (Figure 5B). To verify the reliability of our RNA-seq results, 12 DEGs, including 6 up-regulated and 6 down-regulated genes, were randomly selected and verified by qPCR that their differential expression in NX and LW muscles was consistent with the sequencing results (Figure 5C,D). To decipher the differential processes underlying the neonatal stages of muscle development between LW and NX pigs, we performed GO and KEGG enrichment analyses using the R package clusterProfiler. The top 30 candidates were presented (Figure 5E,F). Notably, the majority of DEGs were enriched within pathways associated with protein folding and protein processing in the endoplasmic reticulum. This robust enrichment strongly underscores the significant disparities in protein synthesis pathways that distinguish the two breeds.

### 3.6. Conjoint Analyses of MeRIP-Seq and RNA-Seq Data

To identify the genes governed by m^6^A modification, we conducted an integrated analysis of MeRIP-seq and RNA-seq data. A total of 8462 peaks showed not only m^6^A modification but also changes in mRNA levels (Figure 6A). In total, 27 genes were picked out with *p* < 0.05 and fold change > 2. Of these selected genes, 11 displayed increased expression along with enhanced m^6^A methylation in LW pigs, whereas 9 genes showed increased expression but reduced m^6^A methylation. Additionally, two genes were down-regulated in LW pigs and had lower m^6^A methylation levels, whereas five genes exhibited decreased expression with increased m^6^A methylation (Figure 6B). The heatmap displays all genes (Figure 6C). We randomly selected five genes for methylated mRNA immunoprecipitation using m^6^A antibody followed by quantitative RCR (MeRIP-qPCR). The results showed that all selected genes exhibited significantly higher expression in the IP group compared with the IgG group (Figure 6D). The mRNA expression levels were also consistently verified in NX and LW pig muscles (Figure 6E), thereby substantiating the reliability of our analysis. We used integrative genomics viewer (IGV) visualisation analysis to provide a more intuitive presentation of the sequencing data. For instance, the SH3BP4 gene exhibited a markedly higher m^6^A binding peak in the 3′UTR region of LW pigs (Figure 6F).

### 3.7. m^6^A Modification Regulated Differential Expression of WFS1 in NX and LW Porcine Muscle

Based on previous studies highlighting the regulatory role of WFS1 (Wolfram syndrome 1) in muscle development, we focused on WFS1. As illustrated by the IGV analysis, a significantly higher m^6^A enrichment was observed in the last exon of WFS1 in NX pigs, with a markedly lower expression level (Figure 7A). Significant enrichment of m^6^A modifications in WFS1 was confirmed by MeRIP-qPCR (Figure 7B). Simultaneously, we validated the high expression of WFS1 in LW pig muscle using quantitative PCR (Figure 7C). We treated PSCs with FB23-2 (2.6 μM) to inhibit FTO expression, thereby increasing global m^6^A modification levels. The results revealed an increase in WFS1 expression (Figure 7D). To investigate whether m^6^A modification affects the stability of WFS1 mRNA, we examined the remaining levels of WFS1 mRNA at different time points after treatment with actinomycin D to assess changes in its half-life. The results showed a significant reduction in the half-life in the FB23-2 treated group (Figure 7E). Because YTHDF2 serves as a major binding protein for m^6^A modification to promote mRNA degradation, we interfered with YTHDF2 while inhibiting FTO. YTHDF2 knockdown alleviated the decrease in WFS1 mRNA levels in cells treated with FB23-2 (Figure 7F). Collectively, these results indicate that m^6^A modification accelerates the degradation of WFS1 in a YTHDF2-dependent manner.

To investigate the reasons for the differences in m^6^A modifications of WFS1 in NX and LW pigs, we performed precise sequencing of the binding peak sequence of WFS1 in the three NX and three LW pigs. To explore the causes behind the disparities in m^6^A modifications of the WFS1 gene between NX and LW pigs, we performed Sanger sequencing of the binding peak sequences of WFS1 in the three NX and three LW pigs to assess sequence differences. Interestingly, sequencing results showed C or heterozygotes of C/T at locus +21,551 (chr8:4363723) of WFS1 in NX pigs, whereas it was a homozygote of T in LW pigs (Figure 7G). The analysis revealed that this SNP was the C site of a conserved m^6^A motif (RRACH), which was absent when the C at 21,551 was mutated to T. To verify whether the SNP site affects the expression of WFS1, we cloned 30 bp sequences upstream and downstream of the SNP site into the pmirGLO vector, as depicted in the upper part of Figure 7H. We transfected plasmids containing either the C site (pmirGLO-MutC), the T site (pmirGLO-MutT), or the empty vector (pmirGLO-WT) into PSCs. After transfection, the cells were treated with FB23-2 (2.6 μM) for 30 h, and normalised luciferase activity was measured. The results are shown in Figure 7H. The relative fluorescence activity of the control pmirGLO vector (pmirGLO-WT) did not change before or after FB23-2 treatment, indicating that m^6^A modification did not affect the fluorescence activity of the pmirGLO vector. The significant difference in normalised luciferase activity intensity between the Mut-C (pmirGLO-MutC) and Mut-T (pmirGLO-MutC) sites suggests that the SNP site markedly influences the generation of m^6^A modifications (Figure 7H). In conclusion, we identified the SNP site C21551T in the WFS1 genes of NX and LW pigs, leading to differences in m^6^A modification, thus affecting WFS1 expression.

## 4. Discussion

The NX pig, a typical obese-type pig breed indigenous to China, possesses high intramuscular fat, large muscle fibre spacing, and sparse muscle fibre arrangement [27,28]. These typical features are not only manifested in adult pigs but were also significantly observed in our results for the neonatal period. This suggests that muscle development in obese- and lean-type pigs shows dichotomic disparity at the early stages of postnatal life. This observation was further reinforced by the sequencing results that identified 1208 DEGs in the muscles of NX and LW pigs. Based on GO and KEGG analyses of the DEGs, we found that most DEGs were enriched in protein folding and protein processing in the endoplasmic reticulum pathways. These findings provide new insights into the differences in muscle development between NX and LW pigs, suggesting that the regulation of protein synthesis may play a significant role. Muscle mass is driven by the balance between protein synthesis and degradation [29]. We propose that the differences in protein synthesis rates between NX and LW pigs contribute to the varying rates of muscle growth.

Recent research into how m^6^A modifications regulate muscle development has shown that both m^6^A hypermethylation and hypomethylation is essential for muscle development. For example, the m^6^A methyltransferase METTL3 plays a crucial role in regulating muscle growth and maintenance in mice, whereas the conditional deletion of METTL3 causes spontaneous muscle wasting over time [22]. In contrast, the expression of the m^6^A demethylase FTO is required for myogenic differentiation in mice, and its conditional knockdown results in a significant reduction in myofibril size, an enlarged interval gap, and a decrease in myofibril number [21]. Our study confirmed that the global reduction in m^6^A significantly promoted the myogenic differentiation of PSCs. We speculate that methylation and demethylation of m^6^A act on different groups of genes, indicating the complexity of m^6^A’s regulation of muscle development. Currently, known pathways regulated by m^6^A modification include the p38 MAPK signalling pathway, mTOR-PGC1α pathway, and MSTN-SMAD3 pathway, etc. [21,22]. Our sequencing results indicate that the Wnt, MAPK, and P13K-Akt signalling pathways may also be regulated by m^6^A.

Regulation of mRNAs by m^6^A modifications, including transcription, splicing, nuclear export, degradation, and maturation, has been widely studied. Post-transcriptional regulation in the cytoplasm is primarily achieved by influencing mRNA stability. The m^6^A binding proteins involved in mRNA stability are YTHDF2, YTHDF3, YTHDC1, YTHDC2, and IGF2BP [30]. The role of m^6^A modification in promoting mRNA degradation is predominantly mediated by YTHDF2. The C-terminal domain of this protein selectively binds to mRNAs harbouring m^6^A, whereas its N-terminal domain is responsible for targeting the YTHDF2-mRNA complex to cellular sites where RNA decay occurs [19]. In our study, half-life assays confirmed that the methylation of m^6^A WFS1 reduces the stability of its mRNA, and we verified that YTHDF2 mediates this process using an effective experimental method.

Recent studies have shown that WFS1 acts as a muscle-specific envelope transmembrane protein, optimizing myogenic gene expression by physically recruiting genes to the nuclear periphery and enhancing their repression during myogenic differentiation. This direct manipulation of gene positions is energy efficient and effective, contributing one-third to two-thirds of the normal repression of a gene during myogenesis [31]. In WFS1 mutant or knockout mouse disease models, obvious growth retardation and significant weight loss has been observed [32,33]. Here, in our study, the expression level of WFS1 was shown to be significantly higher in LW pigs than in NX pigs, which we believe partly explains the higher muscle differentiation potential that LW pigs have during the neonatal period. Furthermore, our research has revealed that m^6^A modification regulates the differential expression of WFS1 between LW and NX pigs. A significant m^6^A-binding peak in the last exon accelerates the degradation of WFS1 in a YTHDF2-dependent manner. Interestingly, an SNP site (C21551T) found in the sequence of the binding peak, as the C site of the m^6^A conserved binding sequence (RRACH), significantly affects WFS1 expression but not protein structure, as the synonymous mutation from GAT to GAC. This SNP site could potentially act as a valuable genetic marker for enhancing meat production traits.

Notably, WFS1 is widely known for its mutation, which causes Wolfram syndrome (WS), a rare genetic disorder most frequently characterised by diabetes insipidus, diabetes mellitus, optic atrophy, and deafness [34,35]. However, the complex underlying pathogenic mechanisms remain unclear. Our findings provide novel insights into the pathogenesis of WS.

## 5. Conclusions

The findings of this study introduce a novel regulatory mechanism for the WFS1 gene, revealing that m^6^A modification affects the degradation rate of WFS1 mRNA. Furthermore, we identified an SNP within the WFS1 gene that affects the occurrence of m^6^A modification. This SNP site could serve as a valuable genetic marker for enhancing meat production traits, while also offering potential insights into the underlying mechanisms of Wolfram syndrome caused by WFS1 mutations.

## Figures and Tables

**Figure 1 cells-13-01744-f001:**
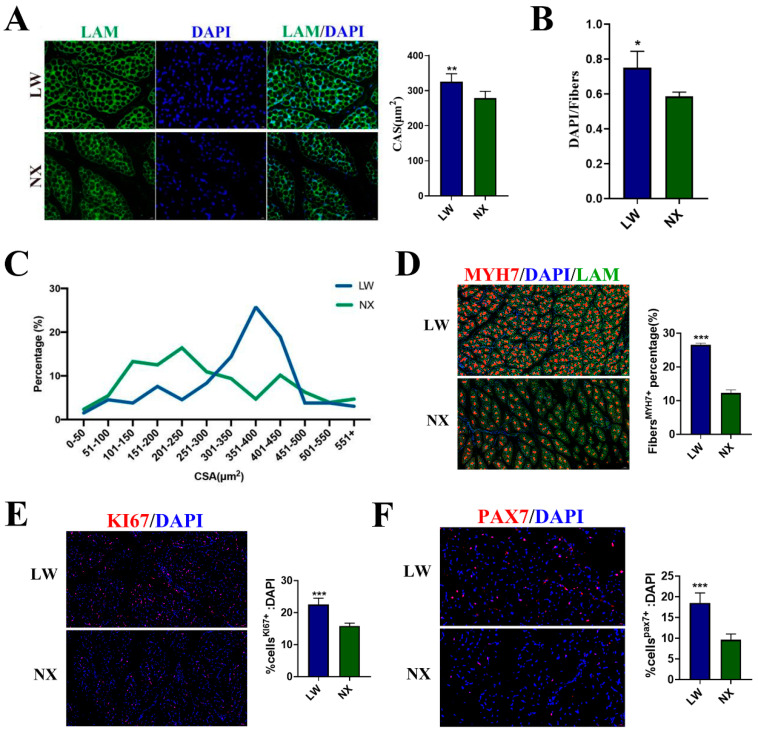
Characteristics of longissimus dorsi in 5–day–old NX and LW pigs. (**A**) Representative LAM/laminin immunostaining (green) of longissimus dorsi sections from NX and LW pigs; DAPI was used for nuclear staining (blue) (scale bar: 100 μm). Mean cross-sectional area (CSA) was counted separately from three random fields. (**B**) The ratio of cell nucleus number to muscle fibre number was calculated from three random fields (n = 3). (**C**) Fibre CSA distribution was counted from four random fields (n = 4). (**D**) Representative images of MYH7 (red) immunofluorescent staining. Scale bar: 100 μm. LAM/laminin was used as a fibre outline (green) (scale bar: 50 µm). The percentage was counted separately from three random fields. (**E**) PAX7 immunostaining (red). Percentage of PAX7+ cells on total DAPI nuclear counterstaining (blue) was provided (scale bar: 20 µm). (**F**) KI67 immunostaining (red). Percentage of KI67+ cells on total DAPI nuclear counterstaining (blue) was provided (scale bar: 50 µm). The ratio was counted separately from three random fields. * *p* < 0.05, ** *p* < 0.01, *** *p* < 0.001.

**Figure 2 cells-13-01744-f002:**
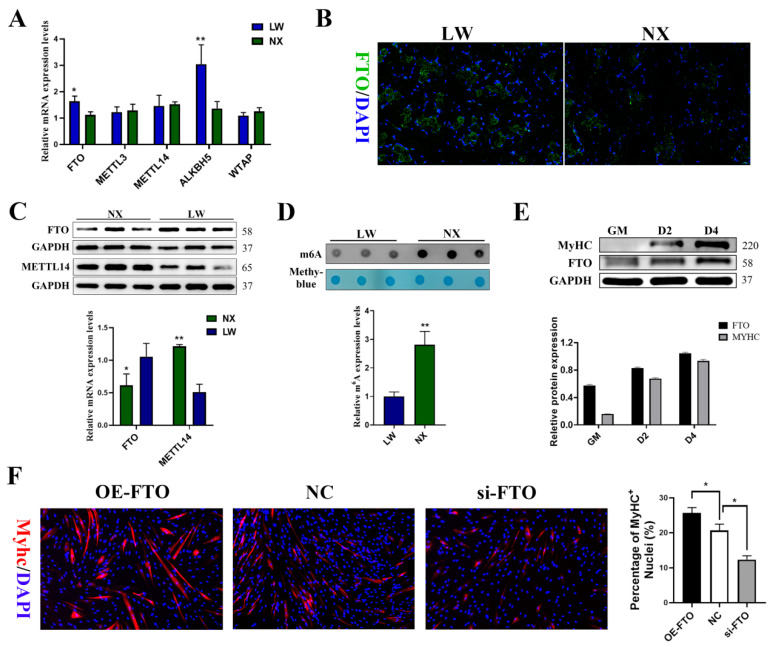
m^6^A modifications are involved in the regulation of muscle development. (**A**) qPCR experiment detected the expression level of m^6^A-related genes in muscle of NX and LW pigs. (**B**) Representative FTO (green) of longissimus dorsi sections from NX and LW pigs; DAPI was used for nuclear staining (blue) (scale bar: 100 μm). (**C**) Western blotting result showed elevated levels of FTO expression and decreased levels of METTL4 expression in LW pig muscle. (**D**) Quantification of m^6^A levels was performed with ImageJ. (**E**) Western blotting showed that the protein expression of FTO was significantly escalated in porcine satellite cells (PSCs) on days 0, 2, and 4 post-differentiation. (**F**) Immunofluorescence staining of Myhc showed that the proportion of Myhc+ cells was significantly increased by FTO overexpression and significantly decreased by FTO knockdown (n = 3). * *p* < 0.05, ** *p* < 0.01.

**Figure 3 cells-13-01744-f003:**
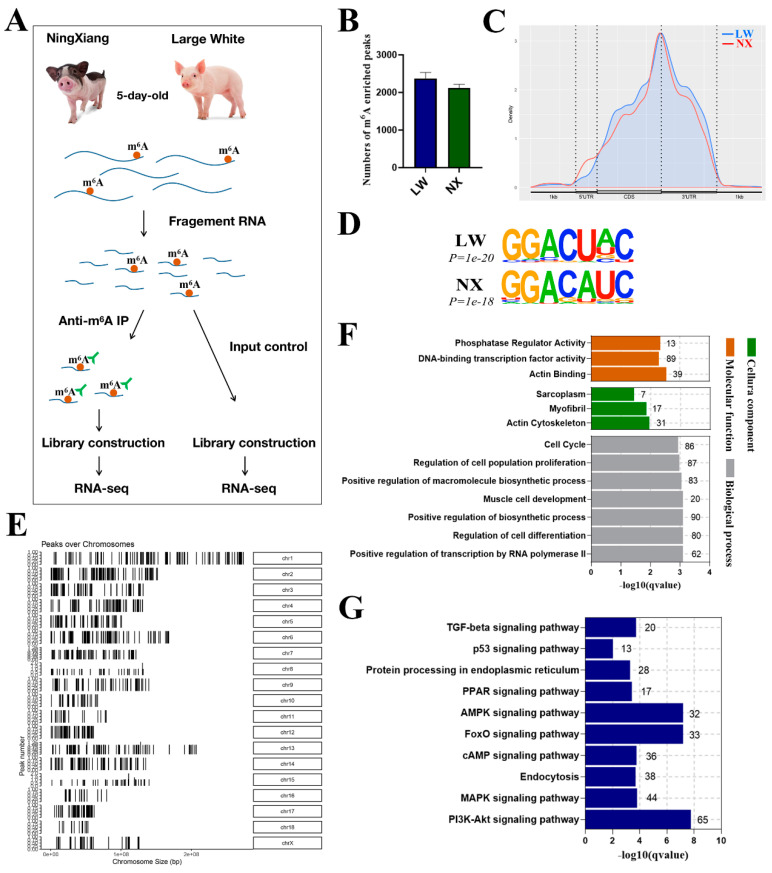
Global features of mRNA m^6^A methylation in LW and NX pig muscles. (**A**) Workflow for detection of m^6^A modifications in NX and LW pig muscle. m^6^A-containing mRNA fragments were purified via immunoprecipitation (IP). (**B**) The average number of m^6^A peaks in each group. (**C**) Density of m^6^A peaks along transcripts. (**D**) The enriched consensus motif of m^6^As in LW and NX pig muscles. (**E**) The distribution of m^6^A enriched regions on the chromosomes of NX−1 pig. GO (**F**) and KEGG (**G**) enrichment analysis in 2035 genes with m^6^A modification in NX−1 pig muscle samples.

**Figure 4 cells-13-01744-f004:**
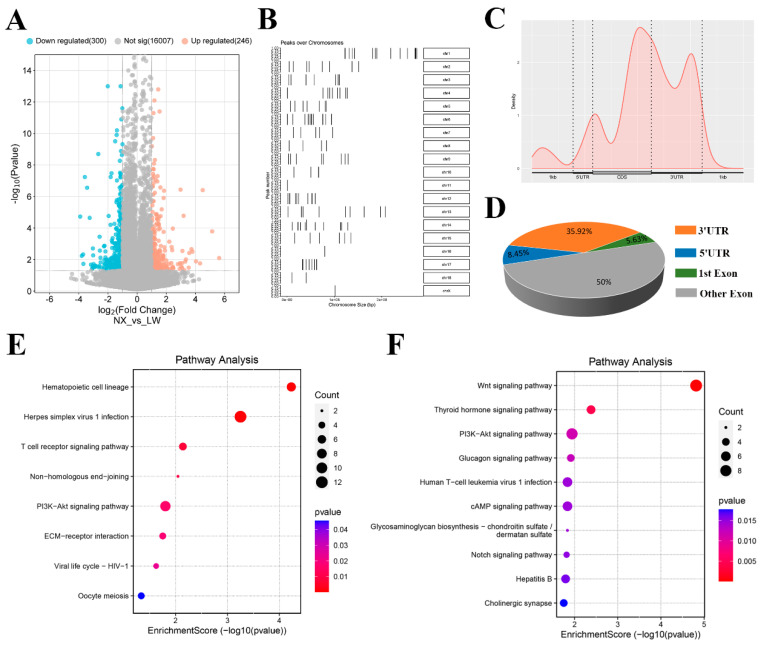
Comparison of differential m^6^A modification between LW and NX pig muscle. (**A**) Volcano plot representation of the differential m^6^A modification genes between LW and NX pig muscle. (**B**) Distribution of differential m^6^A modified genes enriched regions on chromosomes. (**C**,**D**) Distribution of significantly different m^6^A peaks among mRNA regions. KEGG enrichment analysis of differentially up-regulated (**E**) and down-regulated (**F**) m^6^A modification genes in LW compared with NX pig muscle.

**Figure 5 cells-13-01744-f005:**
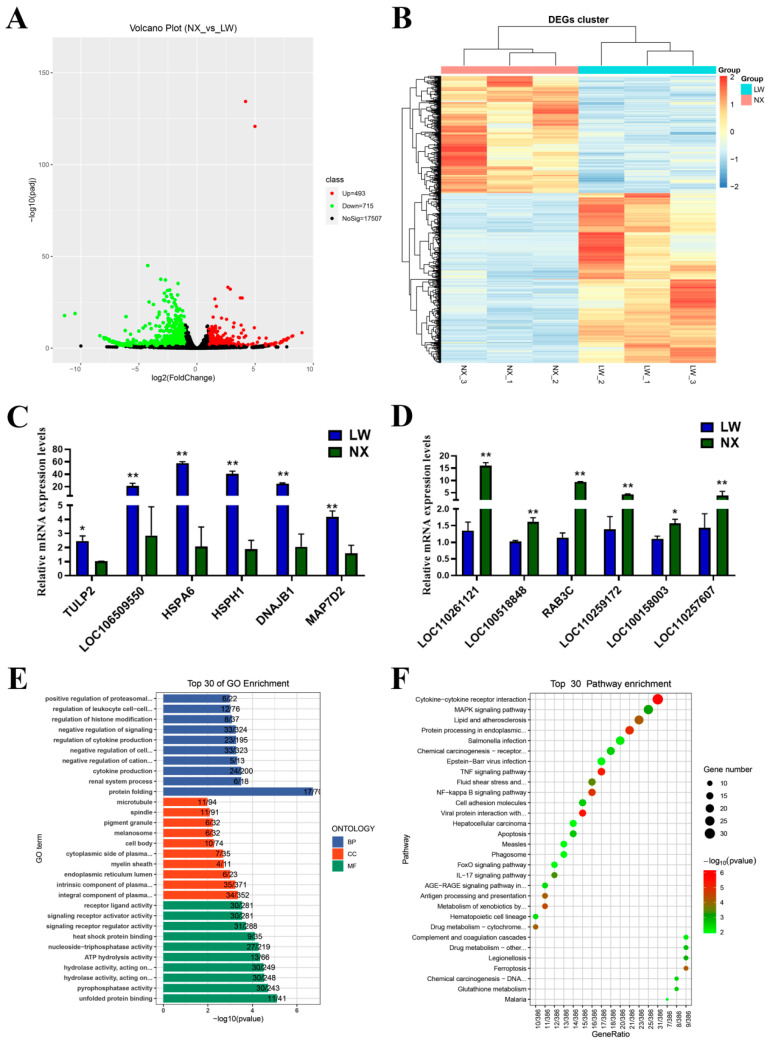
The description of differentially expressed mRNAs in LW and NX pig skeletal muscles. (**A**) Volcano plot representation of the differentially expressed mRNA genes. (**B**) Hierarchical cluster analysis of differentially expressed mRNA genes. (**C**) qPCR result showing the up-regulated mRNA expressions in LW pigs. (**D**) qPCR result showing the down-regulated mRNA expressions in LW pigs. (**E**) Top 30 enriched pathways of differentially expressed genes by GO analysis. (**F**) Top 30 enriched pathways of differentially expressed genes by KEGG prediction. * *p* < 0.05, ** *p* < 0.01.

**Figure 6 cells-13-01744-f006:**
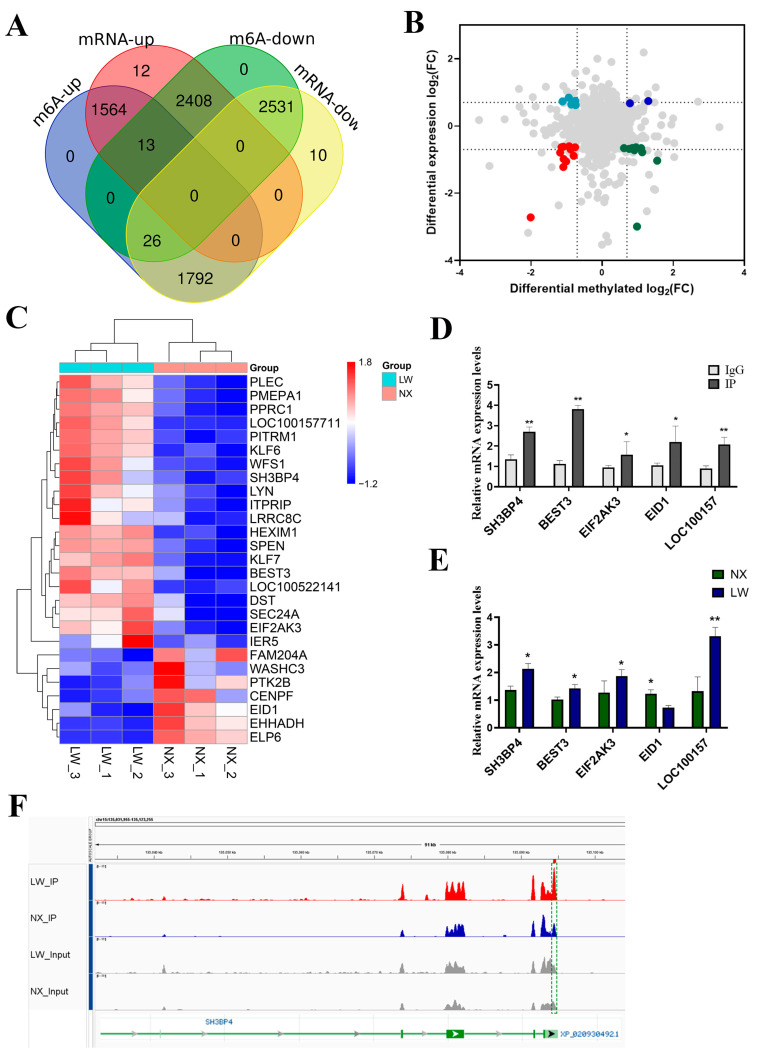
Integrated analysis of RNA-Seq and MeRIP-Seq. (**A**) Venn diagram depicting peak numbers with differential m^6^A methylation and changes in mRNA expression. (**B**) Four-quadrant plot illustrating the distribution of genes. Different colors indicate different changes in RNA expressions or m^6^A methylation levels. (**C**) Heatmap displaying the expression of 27 genes with notable differences in methylation levels. (**D**) MeRIP-qPCR validation of 5 m^6^A-methylated genes, using IgG as a negative control. (**E**) qPCR analysis showing increased expression of 5 hyper-up genes in LW pigs compared with NX pigs. (**F**) IGV visualisation of RNA-seq (grey) data and MeRIP-seq (red and blue) data at the SH3BP4 gene loci. * *p* < 0.05, ** *p* < 0.01.

**Figure 7 cells-13-01744-f007:**
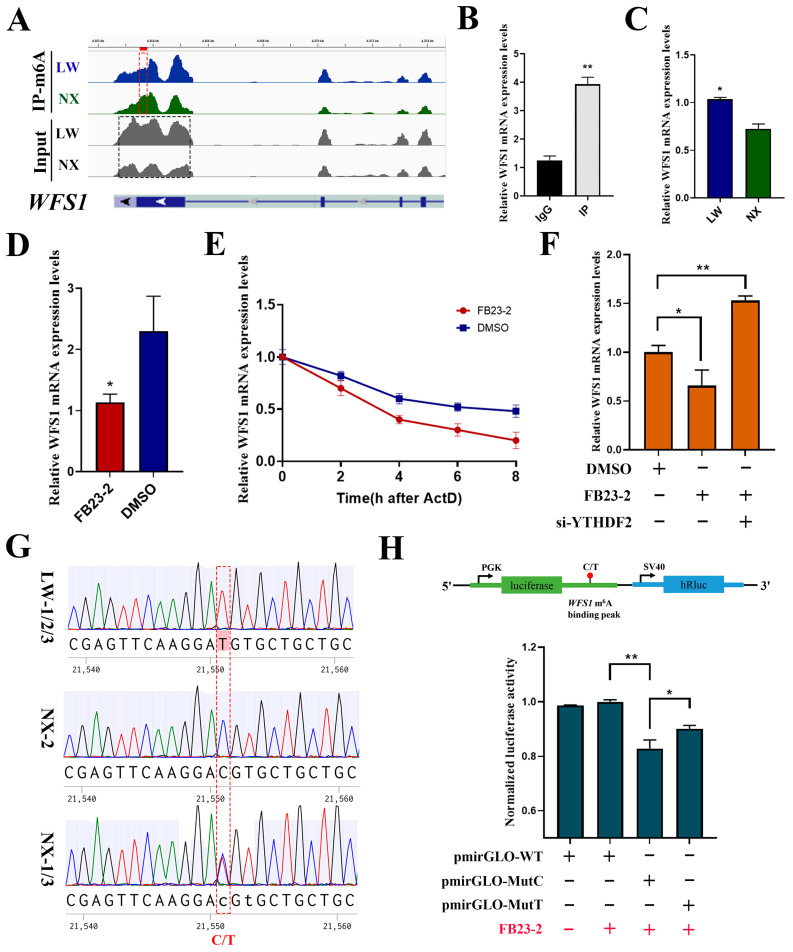
m^6^A modification regulated differential expression of WFS1 in NX and LW pig muscle. (**A**) Genome browser tracks showing RNA-seq (grey) and MeRIP-seq (blue and green) data at WFS1 gene loci. (**B**) MeRIP-qPCR validated the m^6^A modification of WFS1. IgG was used as negative control. (**C**) Detection of WFS1 expression in LW and NX pig muscle. (**D**) RT-PCR analysis of WFS1 mRNA expression in PSCs treated with FB23-2 or DMSO. (**E**) Detection of mRNA degradation rate of WFS1 in PSCs treated with FB23-2 or DMSO. (**F**) Detection of WFS1 expression in DMSO- and FB23-2-treated groups after interference with YTHDF2. (**G**) Sequencing results of these six pigs at the m^6^A binding peak sequence and the SNP site display. (**H**) Dual fluorescence detection of SNP sites on m^6^A binding peaks. The binding fragments containing C or T were cloned into the pmirGLO plasmid, named pmirGLO-MutC and pmirGLO-MutT, respectively (up). Then, they were transfected into PSCs along with the negative control empty vector (pmirGLO-WT), followed by treatment with FB23-2 (red +/−). * *p* < 0.05, ** *p* < 0.01.

## Data Availability

The primary data from this study are available through Gene Expression Omnibus submission GSE266121.

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
