# Peer review of "N6-Methyladenosine RNA Modification Regulates the Differential Muscle Development in Large White and Ningxiang Pigs"

_cells, 2024, doi:10.3390/cells13201744_

Round 1
Reviewer 1 Report
Comments and Suggestions for Authors
This study compares Ningxiang pigs with Large White pigs through physical morphology analysis, N6-Methyladenosine (m6A) data, and RNA expression profiling. The authors identify a single nucleotide polymorphism (SNP) that influences the intensity of m6A modifications between these two pig breeds. While the research topic and experimental design are strong, the scientific writing could be significantly improved for clarity and precision. Here are my comments:
The writing is too vague and needs greater clarity and accuracy
The methods are lacking and need more detail.
In lines 202-204, the word "fibres" should be changed to "fibers" for consistency. Additionally, more specific data regarding the ratio of cell nuclei to muscle fibers for NX and LW pigs should be provided, rather than just stating that the ratio was significantly lower in NX pigs.
In lines 208-209, the phrase "more abundant" regarding PAX7+ and Ki67+ cells is too vague. It's better to include the exact numbers or percentages of these cells in LW pigs compared to NX pigs for clarity.
In lines 260-261, the paper mentions identifying 1208 differentially expressed genes (DEGs) without explaining the methods used for this analysis. The authors should specify the tools, thresholds (such as fold change), and any statistical correction methods (such as FDR) applied in the DEG analysis.
In line 268, when referring to the GO and KEGG databases, it would be helpful to provide the tools or web links used for the analysis, along with the version of the databases.
In line 275, the process for identifying and analyzing m6A modification peaks needs more detail. The authors should specify the software and methods used, including the criteria for calling peaks and filtering data.
In lines 308-309, the reference to "sequencing analysis of the binding peak sequence" is incomplete and unclear. More explanation is needed on what this analysis involved and the methods used.
In line 311, the phrase "sequencing results showed C or heterozygotes of C/T at locus 21,551" should be more specific. The exact genome location should be given in a standard format, such as chrX:21,551 or relative to the start of the gene (e.g., +21,551 or -21,551).
In line 316, regarding the pmirGLO vector, and in line 319, regarding FB23-2 treatment, more details are needed about how these were used in the experiment. This includes information on concentrations, treatment times, and any controls used.
The figure legends, particularly for Figure 7H, need more detailed explanations. It would be useful to describe the context of the figures and explain how they support the paper’s findings
In lines 321-323, it’s unclear how "normalized luciferase activity intensity" directly relates to m6A modification. The authors should clarify the connection between the luciferase assay results, m6A modification, and YTHDF2 binding. Additionally, more explanation is needed about how YTHDF2 binding affects WFS1 mRNA expression and the significance of the differences between Mut-C and Mut-T sites.
Author Response
Comments 1: The writing is too vague and needs greater clarity and accuracy. The methods are lacking and need more detail.
Response 1:Thank you for your valuable feedback. We have taken your comments into consideration and made the necessary additions to clarify any vague sections. In particular, we have provided greater detail in the methods section, including specific data analysis methods to enhance the clarity and accuracy of our work. We appreciate your guidance in improving the manuscript.
Comments 2: In lines 202-204, the word "fibres" should be changed to "fibers" for consistency. Additionally, more specific data regarding the ratio of cell nuclei to muscle fibers for NX and LW pigs should be provided, rather than just stating that the ratio was significantly lower in NX pigs.
Response 2: Thank you for your valuable feedback. We have revised the word 'fibres' to 'fibers' for consistency. Additionally, we updated the section on the presence of oxidative muscle fibers with the precise proportions for LW (26.5%) and NX pigs (12.3%) as requested.
Comments 3: In lines 208-209, the phrase "more abundant" regarding PAX7+ and Ki67+ cells is too vague. It's better to include the exact numbers or percentages of these cells in LW pigs compared to NX pigs for clarity.
Response 3: Thank you for your feedback. We have revised the phrase 'more abundant' to provide the exact percentages of PAX7+ and Ki67+ cells in LW and NX pigs for clarity. Specifically, Ki67+ cells were at 22.5% in LW pigs and 15.86% in NX pigs, while PAX7+ cells were at 18.5% in LW pigs and 9.6% in NX pigs, as requested.
Comments 4: In lines 260-261, the paper mentions identifying 1208 differentially expressed genes (DEGs) without explaining the methods used for this analysis. The authors should specify the tools, thresholds (such as fold change), and any statistical correction methods (such as FDR) applied in the DEG analysis.
Response 4: Thank you for your helpful suggestion. We have specified the methods used for differential gene expression analysis in the Materials and Methods section (2.5). The differentially expressed genes were identified using edgeR software, with DEGs defined by an adjusted P-value (FDR) less than 0.05 and an absolute log2(fold change) greater than 1.
Comments 5: In line 268, when referring to the GO and KEGG databases, it would be helpful to provide the tools or web links used for the analysis, along with the version of the databases.
Response 5: Thank you for your insightful comment. We have revised the manuscript to specify that we used the clusterProfiler package for GO and KEGG enrichment analyses.
Comments 6: In line 275, the process for identifying and analyzing m6A modification peaks needs more detail. The authors should specify the software and methods used, including the criteria for calling peaks and filtering data.
Response 6: Thank you for your valuable suggestion. We have added a detailed description of the process for identifying and analyzing m6A modification peaks in the Materials and Methods section (2.6), including the specific software and R packages used, as well as the criteria for calling peaks and filtering data.
Comments 7: In lines 308-309, the reference to "sequencing analysis of the binding peak sequence" is incomplete and unclear. More explanation is needed on what this analysis involved and the methods used.
Response 7: Thank you for your helpful comment. We realize the wording in this section may have caused some confusion. To clarify, we performed Sanger sequencing to analyze sequence differences in the binding peak regions of the WFS1 gene between NX and LW pigs. We have revised the manuscript to include a more detailed explanation of the methods used.
Comments 8: In line 311, the phrase "sequencing results showed C or heterozygotes of C/T at locus 21,551" should be more specific. The exact genome location should be given in a standard format, such as chrX:21,551 or relative to the start of the gene (e.g., +21,551 or -21,551).
Response 8: Thank you for your valuable suggestion. We have revised the text to provide the exact genome location in a standard format, specifying that the locus is in exon 9 at position +21,551 of the WFS1 gene (chr8:4,363,723).
Comments 9: In line 316, regarding the pmirGLO vector, and in line 319, regarding FB23-2 treatment, more details are needed about how these were used in the experiment. This includes information on concentrations, treatment times, and any controls used.
Response 9: Thank you for your insightful suggestion. We have expanded the text to include details about the plasmids transfected into PSCs: pmirGLO-MutC, pmirGLO-MutT, and pmirGLO-WT (empty vector). After transfection, we treated the cells with FB23-2 (2.6 μM) for 30 hours and measured the normalized luciferase activity. These details have been incorporated into the revised manuscript.
Comments 10:The figure legends, particularly for Figure 7H, need more detailed explanations. It would be useful to describe the context of the figures and explain how they support the paper’s findings.
Response 10: Thank you for your suggestion. We have provided more detailed explanations in the figure legends to better describe the context of the figures.
Comments 11: In lines 321-323, it’s unclear how "normalized luciferase activity intensity" directly relates to m6A modification. The authors should clarify the connection between the luciferase assay results, m6A modification, and YTHDF2 binding. Additionally, more explanation is needed about how YTHDF2 binding affects WFS1 mRNA expression and the significance of the differences between Mut-C and Mut-T sites.
Response 11: Thank you for your insightful comments. We apologize for the previous unclear wording. We have revised the description of the results and conclusions to enhance clarity.
Reviewer 2 Report
Comments and Suggestions for Authors
Dear authors,
thank you for the opportunity to review your manuscript. The text is well-written and requires only minor revisions. Please address the following inaccuracies and suggestions for improvement:
Please include individual data points in the column graphs for better transparency. This is mandatory.
Improve the legend descriptions to clarify whether the data represents technical replicates or biologically independent replicates (e.g., Fig 1A, 1B).
Report the type of statistical analysis performed and specify which statistical method was used to determine significance (e.g., what p-value thresholds correspond to */**/***).
Provide a precise description of the datasets analyzed in Figures 3F, 3G, 4E, and 4F. Was the data curated? Were any cutoffs applied? Is the enrichment measurement based solely on gene count, or does it also include gene expression levels?
Review th whole text for errors like in line 284 s: “The results showed that All genes were…”.
In the Results section, avoid drawing premature conclusions. Only data description statements are permitted. For example, in line 300: "Furthermore, after treatment with actinomycin D, we detected a significant reduction in the half-life of WFS1 mRNA in FB23-2 treated cells, indicating that increased m6A modification led to a decrease in mRNA stability (Fig. 7E)." Please rephrase to ensure conclusions are reserved for the Discussion section.
Correct the acronym Wilms Tumour-associated protein from "WTP" to WTAP (Wilms Tumor-associated Protein).
Expand on the role of m6A to emphasize that it primarily affects post-transcriptional regulation. This includes mRNA processing, splicing, and translation. Provide a more detailed explanation of its role in these processes.
Extend the discussion on the roles of FTO and ALKBH5 in regulating m6A. Include additional examples beyond spermatogenesis and preadipocyte differentiation, to illustrate their broader roles in other biological processes.
Clearly define key terms when they are introduced for the first time, such as the Ningxiang (NX) pig.
Correct the description of WFS1 (Wolfram Syndrome 1). It is not muscle-specific; rather, it is widely expressed in various tissues, including the pancreas, brain, and heart.
The claim that WFS1 "optimizes myogenic gene expression by physically recruiting genes to the periphery" is not well-supported in the literature. WFS1 is better known for its roles in calcium homeostasis and endoplasmic reticulum stress regulation. Revise or provide stronger evidence to support this claim.
Thanks for your efforts.
Author Response
Comments 1: Please include individual data points in the column graphs for better transparency. This is mandatory.
Response 1: Thank you for your suggestion. Our data are independently obtained, such as through statistical analysis of three random, independent fields, and we have added the necessary clarifications in the legend descriptions. However, for consistency and aesthetic reasons, we have not included individual data points in the column graphs.
Comments 2: Improve the legend descriptions to clarify whether the data represents technical replicates or biologically independent replicates (e.g., Fig 1A, 1B).
Response 2: Thank you for your suggestion. We have added the necessary clarifications regarding the statistical methods, including whether the data represents technical replicates or biologically independent replicates, in the figure legends.
Comments 3: Report the type of statistical analysis performed and specify which statistical method was used to determine significance (e.g., what p-value thresholds correspond to */**/***).
Response 3: Thank you for your valuable feedback. We have added details regarding the type of statistical analysis performed in the Materials and Methods section, specifically in subsections 2.5. Differential Expression Analysis and 2.6. MeRIP-Seq Analysiswhere.
Comments 4: Provide a precise description of the datasets analyzed in Figures 3F, 3G, 4E, and 4F. Was the data curated? Were any cutoffs applied? Is the enrichment measurement based solely on gene count, or does it also include gene expression levels?
Response 4: Thank you for your question. As described in the manuscript, we performed GO and KEGG analyses on all genes without any curation or applied cutoffs. We have displayed the pathways of interest in the figures, with the enrichment measurement based on gene count, and the x-axis (-log10(qvalue)) indicating the significance.
Comments 5: Review the whole text for errors like in line 284 s: “The results showed that All genes were…”.
Response 5: Thank you for your feedback. We have revised the sentence to ensure clarity and corrected similar errors throughout the text.
Comments 6: In the Results section, avoid drawing premature conclusions. Only data description statements are permitted. For example, in line 300: "Furthermore, after treatment with actinomycin D, we detected a significant reduction in the half-life of WFS1 mRNA in FB23-2 treated cells, indicating that increased m6A modification led to a decrease in mRNA stability (Fig. 7E)." Please rephrase to ensure conclusions are reserved for the Discussion section.
Response 6: Thank you for your suggestion. We have revised the writing to ensure that conclusion statements are reserved for the Discussion section, and only data descriptions remain in the Results section.
Comments 7: Correct the acronym Wilms Tumour-associated protein from "WTP" to WTAP (Wilms Tumor-associated Protein).
Response 7: Thank you for pointing this out. We have corrected the acronym from 'WTP' to 'WTAP' throughout the manuscript.
Comments 8: Expand on the role of m6A to emphasize that it primarily affects post-transcriptional regulation. This includes mRNA processing, splicing, and translation. Provide a more detailed explanation of its role in these processes.
Response 8: Thank you for your insightful suggestion. We have expanded the discussion on the role of m6A , as suggested.
Comments 9: Extend the discussion on the roles of FTO and ALKBH5 in regulating m6A. Include additional examples beyond spermatogenesis and preadipocyte differentiation, to illustrate their broader roles in other biological processes.
Response 9: Thank you for your suggestion. We have added the relevant discussion on the roles of FTO and ALKBH5 in regulating m6A in the Introduction, as requested.
Comments 10: Clearly define key terms when they are introduced for the first time, such as the Ningxiang (NX) pig.
Response 10: Thank you for your helpful suggestion. We have revised the manuscript to clearly define key terms when they are introduced for the first time.
Comments 11: Correct the description of WFS1 (Wolfram Syndrome 1). It is not muscle-specific; rather, it is widely expressed in various tissues, including the pancreas, brain, and heart.
Response 11: Thank you for your guidance. We have revised the manuscript to focus solely on WFS1's role in muscle cells, without implying that it is muscle-specific, while acknowledging its broader expression in various tissues.
Comments 12: The claim that WFS1 "optimizes myogenic gene expression by physically recruiting genes to the periphery" is not well-supported in the literature. WFS1 is better known for its roles in calcium homeostasis and endoplasmic reticulum stress regulation. Revise or provide stronger evidence to support this claim.
Response 12: Thank you for your insightful feedback. We acknowledge that WFS1 is well-known for its roles in calcium homeostasis and endoplasmic reticulum stress regulation; however, its involvement in these processes during muscle development remains unclear, which is why we did not focus on this aspect in our discussion. Although there is only one study currently supporting WFS1’s role in physically recruiting genes to the periphery to optimize myogenic gene expression, this finding is significant, and we have strengthened our discussion accordingly. Exploring WFS1’s regulation of muscle development through calcium homeostasis and ER stress will be an important direction for our future research.
Round 2
Reviewer 1 Report
Comments and Suggestions for Authors
The response from the authors is satisfactory. I have no further questions.
Reviewer 2 Report
Comments and Suggestions for Authors
Thank you for considering implementing all my suggestions.